# Comparison of Two Diagnostic Techniques for the *Apis mellifera* Varroatosis: Strengths, Weaknesses and Impact on the Honeybee Health

**DOI:** 10.3390/vetsci9070354

**Published:** 2022-07-13

**Authors:** Roberto Bava, Fabio Castagna, Cristina Carresi, Antonio Cardamone, Giovanni Federico, Paola Roncada, Ernesto Palma, Vincenzo Musella, Domenico Britti

**Affiliations:** 1Department of Health Sciences, University of Catanzaro Magna Græcia, 88100 Catanzaro, Italy; roberto.bava@unicz.it (R.B.); fabiocastagna@unicz.it (F.C.); tony.c@outlook.it (A.C.); roncada@unicz.it (P.R.); musella@unicz.it (V.M.); britti@unicz.it (D.B.); 2Interdepartmental Center Veterinary Service for Human and Animal Health, University of Catanzaro Magna Græcia, CISVetSUA, 88100 Catanzaro, Italy; 3Veterinary Pharmacology Laboratory, Institute of Research for Food Safety and Health IRC-FSH, Department of Health Sciences, University Magna Graecia of Catanzaro, 88100 Catanzaro, Italy; 4Istituto Zooprofilattico Sperimentale del Mezzogiorno, Loc. Catona, 89135 Reggio Calabria, Italy; giovanni.federico@izsmportici.it; 5Nutramed S.c.a.r.l. Complesso Ninì Barbieri, Roccelletta di Borgia, 88021 Catanzaro, Italy

**Keywords:** integrated pest management (IPM), honey bee *(Apis mellifera)*, *Varroa destructor*, CO_2_ injection, powdered sugar roll, honey bee welfare and health

## Abstract

**Simple Summary:**

*Varroa destructor* is the main parasite affecting the health of adult honey bees and their larval forms. This mite negatively influences beekeeping income, as it can cause a decrease in yields and the loss of entire colonies of an apiary. To control this parasite, it is essential to establish the level of infestation through a precise diagnosis. In this work, two techniques used for the diagnosis of *V. destructor* on a bee sample were compared: the sugar roll test and CO_2_ injection. Both techniques were evaluated with the *Varroa EasyCheck* tool. This device is particularly versatile because it allows you to choose between different diagnostic techniques. The results of a comparison show that the sugar roll technique is cheaper and results in greater diagnostic accuracy. Both methods do not have a major impact on the health and welfare of bees compared to the alcohol washing method, which results in the death of the test sample. This study is of particular practical value, as it guides beekeepers in choosing the technique to use in a crucial practice for breeding, which is diagnosis.

**Abstract:**

*Varroa destructor* is the most dangerous pest that poses a threat to honey bee survival. In recent years, increasingly worrying phenomena of drug resistance have occurred to various active ingredients of pharmaceutical formulations used to control this parasitosis. Determining the level of infestation is essential to preventing the inappropriate use and abuse of veterinary medicines, and to choose the most appropriate time for treatment. This comparative study investigates the sensitivity and diagnostic accuracy of two field techniques for diagnosing *V. destructor* infestations in hives. The *EasyCheck* device (*Véto-pharma*) was used in two of its application modes, namely, the sugar roll test and carbon dioxide (CO_2_) injection. The experiments were conducted on 15 samples of 300 bees each taken from the same frame and checked for the presence of mites using standard and modified field techniques in both uncaged and caged queen hive conditions. The results demonstrate that the sugar roll technique is significantly more effective and safer than CO_2_ injection, allowing for a higher accuracy in diagnosing a *V. destructor* infestation. Furthermore, the evaluation of mites present on bees in brood block conditions has proven to be particularly reliable. Considering the number of mites on the filter of the device as an additional step helps to implement the diagnostic accuracy of the CO_2_ injection technique, however, not achieving the efficacy results of the sugar roll.

## 1. Introduction

*Varroa destructor* is an ectoparasitic mite, which is currently the most harmful parasite for adult honey bees and their larval forms [1,2]. The mite’s ability to transmit many viruses makes it a major cause of the alarming global disappearance of hives [3]. Parasitic honey bees often show signs of weight loss, reduced size, shortened life expectancy, behavioral changes and deformed wings [4,5,6]. These symptoms can be traced back to the parasite’s viral vector action. Consequently, highly infested colonies develop a parasitic mite syndrome, called varroosis. On the other hand, low levels of infestation by *V. destructor* give rise to undetectable symptoms [1]. If the infestation levels of the colony are not so high as to lead to these symptoms and a consequent collapse, the infested colony still records a decrease in production and pollination capacity [7]. The economic damage caused by the parasite is, therefore, of considerable magnitude. For these reasons, it is of fundamental importance to deepen the knowledge on this parasitosis. The life cycle of *V. destructor* can be divided into two phases: the dispersal phase and the reproductive phase [8]. The first concerns the female mite, which uses the adult honey bee as a “transport vector” and food source. During this period, the mites are often hidden among the abdominal sternitis of the honey bee, in a position that is difficult to reach. In the dispersal phase, honey bees unintentionally participate in the spread of *V. destructor* mites within a colony and between different colonies. Moreover, the reproductive phase begins when the female mite enters an unsealed brood cell containing a fifth-stage bee larva [9]. Due to the temporary external exposure of the mite, the dispersal phase is often used to diagnose an infestation.

Infestation levels significantly increase each month during the brood-rearing season. It is imperative to intervene and control the parasitic population before the colony’s productivity is undermined or its survival threatened. If mite populations develop undetected, the collapse of the untreated and infested honey bee colonies typically occurs in less than two years [10]. Therefore, the presence of *V. destructor* mites determines the need for regular pharmacological treatments. The treatments have often been carried out without having a precise diagnosis of the parasitic load and, in some cases, when not necessary. This behavior has led to the emergence of worrying phenomena of drug resistance towards many of the pharmacological products on the market [11,12,13,14,15]. Currently, it is recommended to apply an integrated pest management (IPM) approach to control mites. An effective IPM program is based on the combination of chemical and nonchemical methods to reduce the pressure on the environment resulting from the control actions put in place to control a pest population [16]. This scheme includes actions such as the rotation of drugs, the use of biological control and/or natural products [17,18,19]. Furthermore, a prompt diagnosis, when the signs of the parasitosis are not yet visible, is essential to avoid decreased production, excessive colony weakening and loss [11,12].

In light of the above, monitoring the level of a *V. destructor* infestation is a key operation of integrated pest management (IPM) programs. A diagnosis allows to determine the extent of a parasite invasion into honey bee colonies, which in turn facilitates the choice of the appropriate treatment method for infected hives. Moreover, a correct diagnosis allows to verify the efficacy of a pharmacological treatment or to monitor a possible reinfestation [20]. Therefore, the constant monitoring of hives throughout the year and maintaining mite levels below economic thresholds is essential, in order to plan harvesting opportunities in advance.

Different diagnostic methods have been implemented to diagnose this parasitosis. Such methods include the monitoring of natural falls, decapping, use of acaricides and the evaluation of the parasite load on a sample of honey bees. The natural fall is recorded on the bottom board with the use of adhesive substances that retain *V. destructor* fallen from the bodies of honey bees. It is a noninvasive procedure that does not require opening the hive. Conversely, this procedure can underestimate or overestimate the level of infestation in relation to inherited grooming behavior. In addition, it takes several days to settle the parasitic load, which must be related to the number of honey bees and to the size of the brood in the hive [21,22]. Acaricides cause the severe death of the mites, which by falling to the bottom of the hive can be counted. However, this method does not allow a diagnosis useful for prevention [23]. A possible diagnosis can be determined by decapping and examining the drone brood. In this case, it is important to underline that the distribution of mites in the brood varies greatly from frame to frame and between the cells of a given portion of the brood [21]. An alternative approach is to assess the infestation level of a sample of adult bees. Using specific detergents or powders, the mites are removed from the adult honey bees. This is a practical and fast method that allows to obtain an immediate diagnosis [20,22,23,24,25,26]. These diagnostic methods on honey bees are not affected by colony size, beehive architecture or the presence of ants. Therefore, among the aforementioned methods, the determination of the infestation rate on adult bees is an immediate and biologically more relevant operation for assessing the health status of a colony. The term economic threshold (ET) indicates the number of mites found following diagnostic interventions, at which control measures must be initiated to avoid reaching economic injury [16]. A number between ~ 2 and 5 mites/100 adult honey bees represents a generally accepted ET for *V. destructor* [16]. It is recommend to perform a *V. destructor* monitoring assessments at least four times during the year, beginning with the phase of population increase (early spring) [27].

The *Varroa EasyCheck* device (*Véto-pharma*) has recently become commercially available. It allows to check the infestation degree of a sample of adult honey bees by washing them with alcohol, by rolling them in powdered sugar or by injecting carbon dioxide (CO_2_) into the device. Although alcohol washes are more accurate and precise, they result in the death of the bee sample. Instead, the sugar roll test and CO_2_ injection are techniques that do not compromise the vitality of the sample and are respectful of animal welfare.

This study aims to compare the sugar roll test and CO_2_ insufflation on a sample of honey bees in assessing the level of a *V. destructor* infestation in the field, constantly monitoring the well-being and health of bees. Powdered sugar adheres to the parasite’s body, in particular, its ambulacrum, making it difficult for mites to remain on honey bees [28]. On the other hand, a low CO_2_ pressure makes honey bees and mites unconscious; in this way, the anesthetized mites dislocate and fall to the bottom of the device. There are several studies concerning the accuracy and sensitivity of washes compared to the sugar roll technique, while comparative data between sugar roll and injection are relatively few [20,29]. Moreover, to our knowledge, there are no studies that attempt to compare diagnostic techniques with the same diagnostic tool. Therefore, this study exploits the multifunctionality of the *EasyCheck* device (Véto-pharma, Palaiseau, France) to describe the differences in diagnostic sensitivity and accuracy between two techniques, by-passing the variability due to the use of different tools. This choice is pivotal for the study, which aims to contribute to the progressive standardization of the most suitable method, eliminating the confounding variables related to the use of different devices.

## 2. Materials and Methods

### 2.1. Location, Conditions and Timing

The tests took place in the province of Catanzaro (Calabria, Southern Italy). The trials were carried out in two apiaries in April 2022. In order to reduce the variables that could interfere with the diagnostic accuracy of the sugar roll method as much as possible, a period of low nectar import and slightly humid days was preferred for the count. Indeed, high humidity or regurgitated nectar from honey bees can interfere with the measurement, as these conditions contribute to maintaining *V. destructor* attached on the honey bees or on the container, thus, falsifying the test. A total of 30 randomly chosen colonies was sampled, 15 for each apiary. The experimental units were colonies of *Apis mellifera ligustica,* housed in Dadant–Blatt hives, similarly managed and balanced in strength and population. The last treatment they received was oxalic acid, dripped in the absence of brood, which was administered in November. With the start of the experimental trials, the two apiaries were managed differently. In particular, in the first apiary, the diagnostic accuracy of the sugar shake method and the CO_2_ injection was compared on broodless hives, in which the queen was caged for 23 days. Twenty-four days (day 24) before all tests, the queen was located and caged. In the second group, the accuracy and sensitivity of the two methods were compared on brood hives with uncaged queens. During egg-laying seasons, most *V. destructor* are found within brood cells and only a small portion parasitize adult bees. This proportion can vary depending on the quantity of brood, the laying stops or the season. In the first case, a high degree of infestation was artificially simulated, while, in the second case, a degree of infestation normally present in the hive was studied. In this way, the accuracy and precision of the two methods were carefully checked under different experimental conditions.

On day zero, two adult bee samples were taken from the same frame of each hive. The distribution of *V. destructor* on adult honey bees within the colony is not homogeneous and the *V. destructor* load on foragers is different from that present on young honey bees. For this reason, it was preferred to perform the analyses on samples of honey bees coming from the same frame and, therefore, probably having honey bees of the same age and behavioral stage [30,31]. The technical procedure carried out on the same frame allowed to reduce all confounding factors related to the characteristics of the subjects, ensuring a high power of the test that was performed. The honey bees were shaken directly from the frame into a bucket and mixed in order to have a homogeneous sample. From the bucket, the honey bees were then collected in two equal containers (120 milliliters of volume). Each sample was weighed before each diagnosis (1 gr accuracy scale), to ensure an equal number of honey bees in the containers. The weighed samples reached approximately 40 gr of adult honey bees, corresponding to approximately 300 honey bees. Although the manufacturer of the *EasyCheck* device (Véto-pharma, Palaiseau, France) recommends using 200 or 300 honey bees to conduct the diagnosis, it was preferred to collect a number of 300 individuals, which represented an optimal sample, as verified by Lee et al. (2010) [26]. This procedure was repeated 15 times for each technique used in each of the two groups. Afterwards, the samples were processed for the two techniques. Finally, for each hive tested, the two samples were recovered and transferred to the laboratory for soap washing, in order to verify the presence of residual mites.

### 2.2. Powdered Sugar Roll Method

A full tablespoon (approximately 25–30 gr) of powdered sugar was placed inside the transparent bowl of the device. The harvested honey bees were then transferred from the container into the transparent bowl. Subsequently, the white basket of the device was replaced, upside-down, in the transparent bowl. Finally, the yellow lid was screwed on. The newly closed device was gently rolled for one minute to evenly coat all honey bees with powdered sugar and then left to rest for another 3 min. After this time, the lid was removed and the *Varroa EasyCheck* was turned upside down and shaken over a large container with a small amount of water, in order to dissolve the sugar. The device was shaken until no more mites came out. Finally, the parasites were counted. Finally, the honey bees were transferred to another container, cooled in dry ice and transported to the laboratory. The residual number of mites present in the sample was verified by washing the bees in a soapy solution. This allowed to calculate the accuracy of the method by measuring the exact number of *V. destructor* in the sample.

### 2.3. CO_2_ Injection Method

The collected honey bees were moved from the container into the transparent bowl. The white basket of the *EasyCheck* device was returned back upside down in the bowl and the lid was placed on top. Through a small opening between the yellow lid and the transparent container, a small amount of CO_2_ was injected for 3–4 s, until the honey bees stopped moving. Then, the lid was quickly screwed on and the device was placed down for 10 seconds to allow the honey bees to go under anesthesia. The *EasyCheck* was turned upside down and gently shaken for 15 s. In this way, the mites were dislodged from the honey bees and collected in the inner surface of the yellow lid. Once the mites were counted, the device was disassembled and inspected for any mites inside. Finally, the honey bees were transferred to another container, cooled in dry ice and taken to the laboratory to check for residual mites.

### 2.4. Laboratory Washes 

Bee samples taken in the field and already processed for diagnosis with sugar roll or CO_2_ injection were inspected for the presence of residual mites in the laboratory. A soapy solution was prepared according to Pietropaoli et al. (2021) [22]. Briefly, 5 mL of commercial liquid dish soap was added to one liter of water. Each honey bee sample was transferred from the field container into a beaker containing 200 mL of soapy solution and shaken on a magnetic stirrer. The mixer was set at a speed of 900 rpm/min at 20 °C. Then, the solution was stirred for 30 min. Two superimposed sieves were used for washing.

The former had a mesh that did not allow the passage of honey bees, but only mites and water; instead, the second sieve allowed the retention of mites and filtered the washing water. Several washes were carried out and the last was performed with high-pressure water, able to leave no more mites on honey bees. Finally, the number of mites contained in the second sieve was counted.

### 2.5. Statistical Analysis

The percentage of the infestation of each sample was calculated with the following formula [17]: IL = (VN/BN) × 100. IL: infestation level of bees with *V. destructor*; VN: *V. destructor* parasites number found in a sample; BN: bees number in a sample. 

Data were analyzed with GraphPad PRISM 6.0 (GraphPad Software, Inc., La Jolla, CA, USA). Results are shown as mean ± SEM. Normality was tested using D’Agostino Pearson’s test. Normally distributed data were analyzed with one-way ANOVA, followed by Tukey’s test, while data without normal distribution were analyzed using Kruskal–Wallis analysis of variance and subsequent Dunn’s tests or Mann–Whitney test. A *p*-value of < 0.05 was considered significant.

## 3. Results

### 3.1. Evaluation of the Sugar Roll Method Compared to CO_2_ Injection in Noncaged Queen Hive Samples

A first comparison of the number of dispersal mites in bee colonies was performed using sugar roll or CO_2_ injection techniques in uncaged queen hive samples.

The data reported a statistically significant reduction in the number of mites counted with the CO_2_ injection technique compared to that with the sugar roll (Figure 1; * *p* < 0.05). This result led to a significant underestimation of the degree of infestation diagnosed with the CO_2_ injection method compared to that with the sugar roll (Figure 2; * *p* < 0.05).

A further mite count was performed by the authors through an inspection of the device and compared to the first one. Interestingly, this additional step contributed to improving the mite count, leading to a more accurate infestation rate in the CO_2_ injection technique (Figure 1 and Figure 2), significantly ameliorating the efficacy of the method (Figure 3B,D). Contrarily, the inspection of the device did not contribute to the effectiveness of the sugar roll method, by which the highest efficacy rate of the diagnostic method was recorded (Figure 1, Figure 2 and Figure 3A,C).

A soapy wash after the CO_2_ injection led to a significant increase in the number of counted mites and, therefore, in the degree of infestation registered when compared to the CO_2_ injection alone (Figure 1 and Figure 2; ^§^
*p* < 0.05), while the same laboratory wash did not significantly improve the accuracy of *V. destructor* diagnosis after the sugar roll procedure (Figure 1 and Figure 2). Finally, the soapy wash greatly improved the mite count and the accuracy of the *V. destructor* diagnosis, even when compared to the CO_2_ injection followed by a device inspection (Figure 1 and Figure 2; ^çç^
*p* < 0.01 and ^Ω^
*p* < 0.05), while no significance was recorded in the case of the sugar roll method in any of the conditions tested (Figure 1 and Figure 2).

### 3.2. Evaluation of the Sugar Roll Method Compared to CO_2_ Injection in Caged Queen Hive Samples

The data reported a statistically significant reduction in the number of mites counted with the CO_2_ injection compared to that with the sugar roll technique (Figure 4). Additionally, the mite count resulting from the CO_2_ injection followed by a device inspection was significantly lower when compared to the sugar roll count (Figure 4). The counting data obtained led to a statistically significant reduction in the percentage of infestation diagnosed following the CO_2_ injection compared to that diagnosed with the sugar roll technique, with or without the device inspection step (Figure 5). The soapy washing confirmed that the mite count and the diagnosed infestation rate were seriously underestimated in the case of the CO_2_ injection compared to the sugar roll (Figure 4 and Figure 5). The graphs shown in Figure 6 confirm the different degree of diagnostic efficacy observed in the evaluation of the two field methods for the diagnosis of *V. destructor*.

## 4. Discussion

In the present research study, the diagnostic accuracy of the CO_2_ injection and the sugar roll technique were compared using the same device. This particularity eliminated the distortion factors of the result due to the variability of the instrument. Our results clearly showed that the sugar roll technique was significantly more effective than the CO_2_ injection technique, allowing for a higher accuracy in diagnosing the *V. destructor* infestation. Furthermore, the evaluation of mites present on bees in brood block conditions using the sugar roll method was proved to be particularly reliable.

A further mite count, performed through a device inspection, contributed to the improvement of the mite count, leading to a more accurate infestation rate after CO_2_ injection and significantly ameliorating the efficacy of the method. This procedure, intended as an additional step, helped in ensuring the health of bees and in the accuracy of the *V. destructor* infestation diagnosis, underlying the lower effectiveness of the CO_2_ injection technique compared to that of the sugar roll. The soapy washing confirmed that the mite count and the diagnosed infestation rate were seriously underestimated in the case of CO_2_ injection compared to the sugar roll. The data obtained in the present study showed a greater diagnostic accuracy and efficacy of the sugar roll method when compared to the CO_2_ injection, using the same type of device.

The diagnostic accuracy of the sugar roll method could be explained by considering the pushing pressure that the powdered sugar mass exerted on the mites, causing them to fall more effectively. Conversely, after exposure of the bee samples to CO_2_, mites were often found inside the device, precisely in the inner part of the white basket or in the holes or in the transparent bowl. The mites found in the device, not attached to the body of the bees, were not included in the total count of the field collection according to the protocol of *Véto-pharma.* For this reason, in the present work, the additional inspection step of the device was proposed aiming to obtain a more accurate diagnosis of *V. destructor*. Finally, it should be considered that the data could have also been minimally affected by the intrinsic variability of the sample, even if it was minimized by taking each sample from the same brood comb. In this sense, it would be advisable to carry out studies trying to identify the most representative type of sample that best reflected the degree of infestation in order to reduce internal variability.

With particular attention devoted to the well-being of honey bees, it is necessary to underline that both diagnostic methods used were certainly of more ethical use than washing with alcohol, as they did not damage or kill the honey bees examined. Indeed, following a diagnosis of *V. destructor* infestation through these techniques, the honey bee samples could be reintegrated into the hive. Deepening the analysis, it must be stated that the jet of CO_2_ can, in some cases, damage the sample of bees. Indeed, it has been observed that an incorrect flow, directed towards the bees, causes cold burns on the bee’s body, strongly limiting its use. The symptoms are more evident following a longer exposure time. Therefore, greater attention must be placed in the use of this diagnostic method to avoid stressful conditions or the death of the sample bees. It is recommended that the CO2 flow is always correctly directed and the initiation of sedation should be visually assessed and the gas supply stopped shortly before. Therefore, a shorter CO_2_ exposure time could be considered in order not to excessively damage the sample. Such behavior would be beneficial for the health and well-being of the honey bees.

The economic aspect is, nevertheless, important and should be considered. For the injection of carbon dioxide, a specific injector for the *EasyCheck* tool by *Véto-pharma* was used. The device required the use of 16-gram CO_2_ cartridges readily available on the market. With a 16-gram can of CO_2_, we were able to examine 4–5 hives (blowing CO_2_ into the device on average for 3–4 s), while with a 125-gram pack of powdered sugar, we were able to diagnose the infestation of 5–6 hives. Additionally, a pack of powdered sugar has a lower average price than a pack of CO_2_ cartridges. The sugar roll test is, therefore, cheaper and may be most suitable for beekeepers to perform field diagnoses at a lower outlay. A final consideration concerns the viability of the mites. The mites obtained with the sugar roll test are often stressed and sometimes die [17]. Mites collected with the CO_2_ injection recover quickly and are less stressed. Therefore, the parasites collected with CO_2_ injection can be collected in large numbers and used in laboratory studies to assess the efficacy of drugs or pharmacologically active natural substances [17,18] and to evaluate the phenomena of resistance to the active ingredients present in commercially available products.

## 5. Conclusions

The need for an accurate diagnosis is of the utmost importance. In assessing the level of infestations, even two or three mites counted, more or less, could make a difference in the breeder’s choices. In our study, variations were observed between samples from the same colony in different cases. Variations fluctuated between a similar number but, in some cases, even doubled. A further inspection of the instrument could improve the count. In any case, the CO_2_ injection failed to determine a good separation of the mites from the bee’s body compared to the separation determined by the sugar roll test, underestimating the level of infestation.

In conclusion, the CO_2_ method proved less reliable for quantifying V. destructor parasites on adult bees. On the other hand, the sugar roll test provided a reliable value for the number of mites contained in the samples, confirming its efficacy and greatly improving the accuracy of the diagnosis of V. destructor infestations.

However, the need for a better standardization of the V. destructor infestation diagnosis is evident, and should combine the protection of the welfare of bees analyzed with the feasibility, smartness and highest rate of accuracy of the method.

## Figures and Tables

**Figure 1 vetsci-09-00354-f001:**
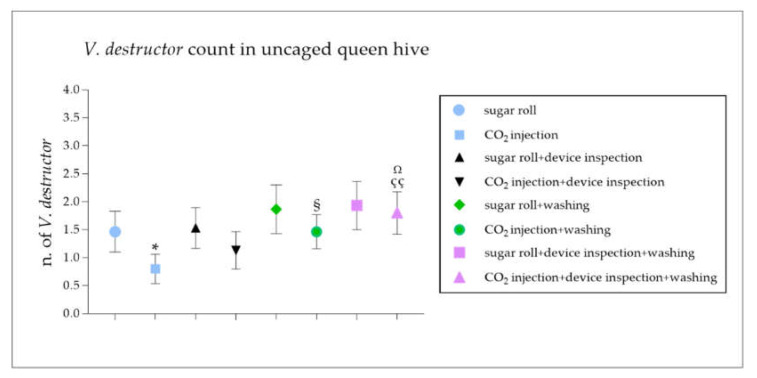
Comparison between the number of dispersal *V. destructor* in colonies of *Apis mellifera ligustica* tested using standard techniques (sugar roll or CO_2_ injection) and modified field techniques (sugar roll + device inspection or CO_2_ injection + device inspection). The graph shows *V. destructor* counts performed with sugar roll or CO_2_ injection techniques, with or without a device inspection, in uncaged queen hive samples. Additionally, *V. destructor* counts derived from sugar roll or CO_2_ injection techniques, with or without a device inspection and/or a water washing were performed. The data are presented as mean ± SEM. * *p* < 0.05 vs. sugar roll, ^§^
*p* < 0.05 vs. CO_2_ injection, ^çç^
*p* < 0.01 vs. CO_2_ injection, ^Ω^
*p* < 0.05 vs. CO_2_ injection and device inspection; Mann–Whitney test (*n* = 300). Data are from 15 different independent counts.

**Figure 2 vetsci-09-00354-f002:**
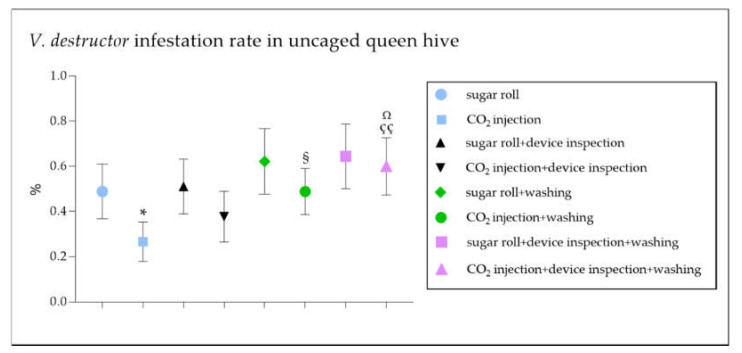
Comparison between *V. destructor* infestation rate in colonies of *Apis mellifera ligustica* tested using standard techniques (sugar roll or CO_2_ injection) and modified field techniques (sugar roll + device inspection or CO_2_ injection + device inspection). The graph shows the percentage of *V. destructor* infestation performed with sugar roll or CO_2_ injection techniques, with or without a device inspection and/or a water washing, in uncaged queen hive samples. The data are presented as mean ± SEM. * *p* < 0.05 vs. sugar roll, ^§^
*p* < 0.05 vs. CO_2_ injection, ^çç^
*p* < 0.01 vs. CO_2_ injection, ^Ω^
*p* < 0.05 vs. CO_2_ injection and device inspection; Mann–Whitney test (*n* = 300). Data are from 15 different independent counts.

**Figure 3 vetsci-09-00354-f003:**
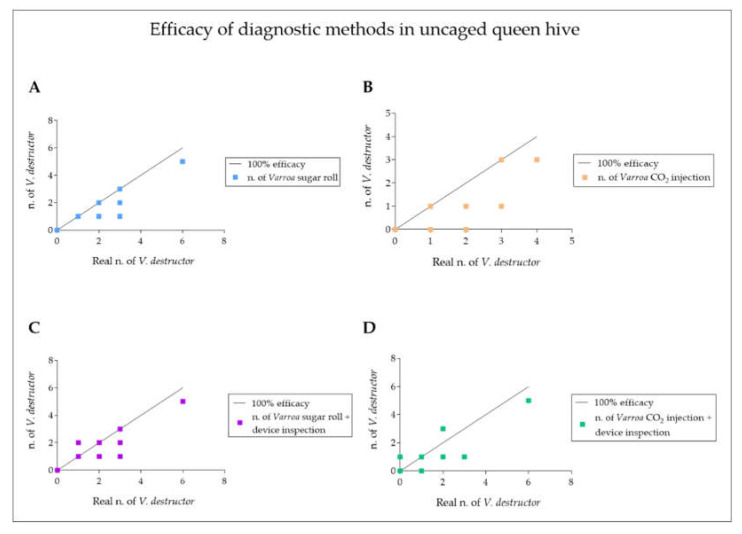
The panel reports the degree of efficacy of the diagnostic methods used to analyze uncaged queen hive samples of *Apis mellifera ligustica***.** The graphs show the number of *V. destructor* counted with sugar roll test (**A**), CO_2_ injection (**B**), sugar roll test and device inspection (**C**) and CO_2_ injection and device inspection (**D**), as a function of their real number in bee samples (*V. destructor* collected via the tested method added to those collected after washing the sample). Each point represents a measurement. The points on the black line represent an efficiency of the measurement method of 100%.

**Figure 4 vetsci-09-00354-f004:**
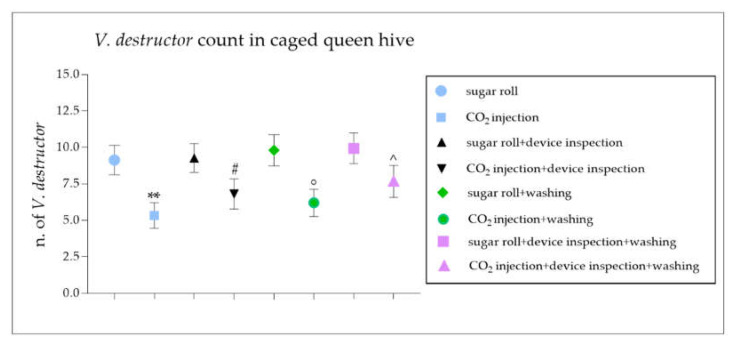
Comparison between the number of dispersal *V. destructor* in colonies of *Apis mellifera ligustica* tested using standard techniques (sugar roll or CO_2_ injection) and modified field techniques (sugar roll and device inspection or CO_2_ injection and device inspection). The graph shows *V. destructor* count performed with sugar roll or CO_2_ injection techniques, with or without a device inspection, in caged queen hive samples. Additionally, *V. destructor* counts derived from sugar roll or CO_2_ injection techniques, with or without a device inspection and/or a water washing were performed. The data are presented as mean ± SEM. ** *p* < 0.01 vs. sugar roll and ^#^
*p* < 0.05 vs. sugar roll and device inspection, ^°^
*p* < 0.05 vs. sugar roll and washing and ^^^
*p* < 0.05 vs. sugar roll and device inspection and washing; Mann–Whitney test (*n* = 300). Data are from 15 different independent counts.

**Figure 5 vetsci-09-00354-f005:**
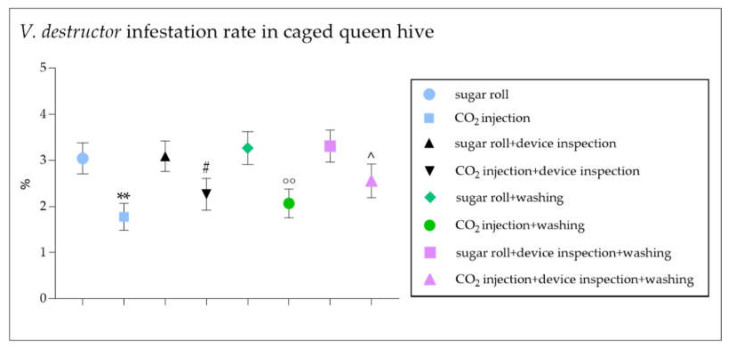
Comparison between *V. destructor* infestation rate in colonies of *Apis mellifera ligustica* tested using standard techniques (sugar roll or CO_2_ injection) and modified field techniques (sugar roll and device inspection or CO_2_ injection and device inspection). The graph shows the percentage of *V. destructor* infestation performed with sugar roll or CO_2_ injection techniques, with or without a device inspection and/or a water washing, in caged queen hive samples. The data are presented as mean ± SEM. ** *p* < 0.01 vs. sugar roll, ^#^
*p* < 0.05 vs. CO_2_ injection, ^°°^
*p* < 0.01 vs. sugar roll and washing and ^^^
*p* < 0.05 vs. sugar roll and device inspection and washing; Mann–Whitney test (*n* = 300). Data are from 15 different independent counts.

**Figure 6 vetsci-09-00354-f006:**
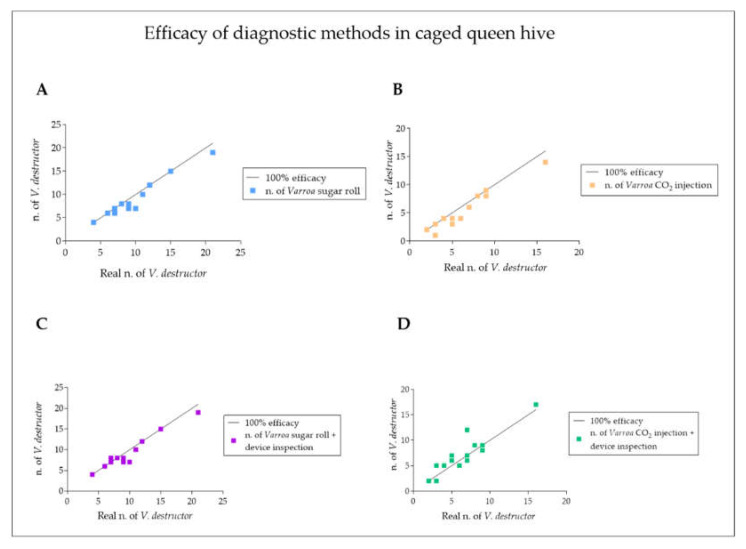
The panel reports the degree of efficacy of the diagnostic methods used to analyze caged queen hive samples of *Apis mellifera ligustica***.** The graphs show the number of *V. destructor* counted with sugar roll test (**A**), CO_2_ injection (**B**), sugar roll test and device inspection (**C**) and CO_2_ injection and device inspection (**D**), as a function of their real number in bee samples (*V. destructor* collected via the tested method added to those collected after washing the sample). Each point represents a measurement. The points on the black line represent an efficiency of the measurement method of 100%.

## Data Availability

Not applicable.

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
