# Peer review of "Comparison of Two Diagnostic Techniques for the Apis mellifera Varroatosis: Strengths, Weaknesses and Impact on the Honeybee Health"

_vetsci, 2022, doi:10.3390/vetsci9070354_

Round 1
Reviewer 1 Report
Varroa destructor is the most dangerous pest that poses a threat to the honey bee survival. This study investigates the sensitivity and diagnostic accuracy of two field techniques for diagnosing Varroa destructor infestation in hives. The results showed that the CO2 method proved less reliable for quantifying Varroa parasites on adult bees. The sugar roll test provided a reliable value for the number of mites contained in the samples confirming its highest safety, efficacy and greatly improving the accuracy of the diagnosis of Varroa infestation. Here I suggest some minor revisions of the text.
1. Authors should carefully check the writing of the manuscript.
2. Line 32, "The results demonstrated..." with an extra space before it
3. line 54, "the reproductive phase begins...", with an extra space.
4. line 342, "injection can be collected..." one more space
5. The resolution of the figures is low, and the figures with high resolution should be provided.
6. In Figure 3, it is better to name each part of the figure, for example figure 6, and explain each one.
7. Figure 6, it is better to explain each figure, including Fig.6A, Fig.6B, Fig.6C and Fig.6D in the figure legend.
8. Figure 3, the number of Varroa parasites is not large. Does it affect the accuracy of the results?
Reviewer 2 Report
This study compares the effectivness of two commercially available diagnostic methods for measuring levels of Varroa destructor infestations. In general, the probem of varroa mite is widespread and thus there is a need of effective diagnostic methods that can help with correct application of treatments and can also help to prevent unctorolled use of drugs and development of drug resistance. Therefore, this study might be of interest for the wide audience. However, there are a few major and a few minor issues that have to be fixed first. Please see line by line comments.
